# ON THE TUNABILITY OF OPTIMIZERS IN DEEP LEARNING

## ABSTRACT

There is no consensus yet on the question whether adaptive gradient methods like Adam are easier to use than non-adaptive optimization methods like SGD. In this work, we fill in the important, yet ambiguous concept of 'ease-of-use' by defining an optimizer's *tunability*: How easy is it to find good hyperparameter configurations using automatic random hyperparameter search? We propose a practical evaluation protocol for optimizer tunability that can form the basis for a fair optimizer benchmark. Evaluating a variety of optimizers on an extensive set of standard datasets and architectures, we find that Adam is the most tunable for the majority of problems, especially with a low budget for hyperparameter tuning.

## 1 INTRODUCTION

With the ubiquity of deep learning in various applications, a multitude of first-order stochastic optimizers (Robbins & Monro, 1951) have been in vogue. They have varying algorithmic components like momentum (Sutskever et al., 2013) and adaptive learning rates (Tieleman & Hinton, 2012; Duchi et al., 2011; Kingma & Ba, 2015). With all these choices, picking the optimizer is among the most important design decisions for machine learning practitioners. For this decision, the best possible generalization performance is certainly an important characteristic to be taken into account. However, we argue that in practice, an even more important characteristic is whether the best possible performance can be reached with the available resources.

The performance of optimizers strongly depends on the choice of hyperparameter values such as the learning rate. In the machine learning research community, the sensitivity of models to hyperparameters has been of great debate recently, where in multiple cases, reported model advances did not stand the test of time because they can be explained by better hyperparameter tuning (Lucic et al., 2018; Melis et al., 2018; Henderson et al., 2018). This has led to calls for using automatic hyperparameter optimization methods with a fixed budget for a fairer comparison of models (Sculley et al., 2018; Feurer & Hutter, 2019; Eggensperger et al., 2019). For industrial applications, automated machine learning (AutoML, Hutter et al., 2019), which has automatic hyperparameter optimization as one of its key concepts, is becoming increasingly more important. In both cases, an optimization algorithm that achieves good performances with relatively little tuning effort is arguably substantially more useful than an optimization algorithm that achieves top performances, but reaches it only with a lot of careful tuning effort. Hence, we advocate that the performance obtained by an optimizer is not only the best performance obtained when using that optimizer, but also has to account for the cost of tuning its hyperparameters to obtain that performance, thus being dichotomous. We term this concept *tunability* in this paper.

Despite the importance of this concept, there is no standard way of measuring tunability. Works that propose optimization techniques show their performance on various tasks as depicted in Table 1. It is apparent that the experimental settings, as well as the network architectures tested, widely vary, hindering a fair comparison. The introduction of benchmarking suites like DEEPOBS (Schneider et al., 2019) have standardized the tested architectures, however, this does not fix the problem of selecting the hyperparameters themselves, and the effort expended in doing so. Previous studies treat tunability to be the best performance obtained on varying a hyperparameter (Schneider et al., 2019) or by measuring the improvement in performance by tuning a hyperparameter (Probst et al., 2019), but do not take any cognizance to the intermediate performance during the tuning process.

Table 1: Experimental settings shown in the original papers of popular optimizers. The large differences in test problems and tuning methods make them difficult to compare. $\gamma$ denotes learning rate, $\mu$ denotes momentum, $\lambda$ is the weight decay coefficient.

| Method | Datasets | Network architecture | Parameter tuning methods |
|---|---|---|---|
| SGD with momentum (Sutskever et al., 2013) | Artificial datasets MNIST | Fully-connected LSTM | $\mu = 0.9$ for first 1000 updates then $\mu \in \{0, 0.9, 0.98, 0.995\}$. other schedules for $\mu$ are used & $log_{10}(\gamma) \in \{-3, -4, -5, -6\}$ |
| Adagrad (Duchi et al., 2011) | ImageNet ranking Reuter RCV1 MNIST KDD Census | Single layer Handcrafted features Histogram features | Perfomance on dev-set |
| Adam (Kingma & Ba, 2015) | IMDb MNIST CIFAR 10 | Logistic regression Multi-layer perceptron Convolutional network | $\beta_1 \in \{0, 0.9\}$ $\beta_2 \in \{0.99, 0.999, 0.9999\}$ $log_{10}(\gamma) \in \{-5, -4, -3, -2, -1\}$ |
| AdamW (Loshchilov & Hutter, 2019) | CIFAR 10 ImageNet 32×32 | ResNet CNN | $log_2(\gamma) \in \{-11, -10 \cdots -1, 0\}$ $log_2(\lambda) \in log_2(10^{-3}) + \{-5, -4, \ldots, 4\}$ |

In this paper, we introduce a fair evaluation protocol for tunability based on automatic hyperparameter optimization, and simple evaluation measures that allow to compare the performance of optimizers under varying resource constraints. By evaluating on a wide range of 9 diverse tasks, we aim to contribute to the debate of adaptive vs. non-adaptive optimizers (Wilson et al., 2017; Shah et al., 2018; Chen & Gu, 2018) . To reach a fair comparison, we experiment with several SGD variants that are often needed to reach good performance. Although a well-tuned SGD variant is able to reach the top performance in some cases, our overall results clearly favor adaptive gradient methods. We therefore conclude that there is substantial value in adaptive gradient methods.

## 2 MEASURING TUNABILITY

Given the dichotomy of the problem of tunability, we argue that it needs to take into account

1. how difficult it is to find a good hyperparameter configuration for the optimizer,
2. the absolute performance of the optimizer.

To see why both are needed, consider Figure 1.a, which shows the performance in terms of loss of four different optimizers as a function of its only hyperparameter $\theta$ (by assumption). If we only consider requirement #1, optimizer C would be considered the best, since every hyperparameter value is the optimum. However, its absolute performance is poor, making it of low practical value. Moreover, due to the same shape, optimizers A and B would be considered equally good, although optimizer A clearly outperforms B. On the other hand, if we only consider requirement #2, optimizers B and D would be considered equally good, although optimizer D's optimum is harder to find.

As we show in Section 5, no existing definition of tunability takes both requirements into account. In the following, we present a formulation that does so.

### 2.1 PRELIMINARIES: HYPERPARAMETER OPTIMIZATION

We define hyperparameter optimization (HPO) (Feurer & Hutter, 2019) as follows:

**Definition.** *Let $\mathcal{M}$ be an optimization algorithm with N hyperparameters $(\theta_1, \ldots, \theta_N) \in \Theta$. Let a specific instantiation of $\mathcal{M}$ with $\boldsymbol{\theta} \in \Theta$ be denoted by $\mathcal{M}_{\boldsymbol{\theta}}$. Thus, given a dataset $D = D_{train} \bigcup D_{val}$, the following objective is minimized*

$$\boldsymbol{\theta}^{\star} = \arg \min_{\boldsymbol{\theta} \in \Theta} \mathcal{L}(\mathcal{M}_{\boldsymbol{\theta}}, D_{val})$$

*where $\mathcal{M}_{\boldsymbol{\theta}}$ is trained on $D_{train}$. In our work, we use $\mathcal{L}$ to be validation loss.*

We use $\mathcal{L}(\boldsymbol{\theta})$ to refer to $\mathcal{L}(\mathcal{M}_{\boldsymbol{\theta}}, D_{val})$ for brevity. In our experiments, we use the time-tested Random Search (Bergstra & Bengio, 2012) algorithm for HPO for simplicity.

## 2.2 DEFINING TUNABILITY

The ability to easily find good minima depends on the loss surface $\mathcal{L}$ itself. Thus, tunability is a characterization of the HPO's loss function $\mathcal{L}$. Here we present a quantification of this idea, which we illustrate with Figure 1.b.

Let us assume that there are two optimizers E & F, both with hyperparameter $\theta$, both of them used to minimize a function (e.g. train a neural network). Let the loss functions of HPO be $\mathcal{L}_E$ and $\mathcal{L}_F$ respectively. As the figure shows, the minimum of $\mathcal{L}_E$ is lower than that of $\mathcal{L}_F$ (denoted by $\theta_E^\star$ and $\theta_F^\star$) i.e. $\mathcal{L}_E(\theta_E^\star) < \mathcal{L}_F(\theta_F^\star)$. However, the minimum of $\mathcal{L}_E$ is much sharper than that of $\mathcal{L}_F$, and

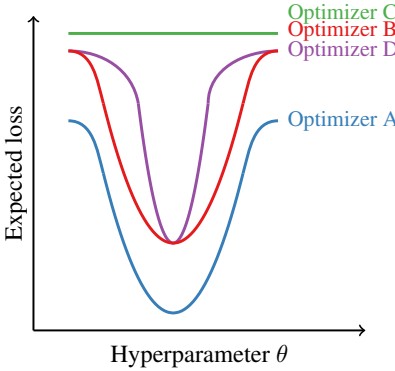

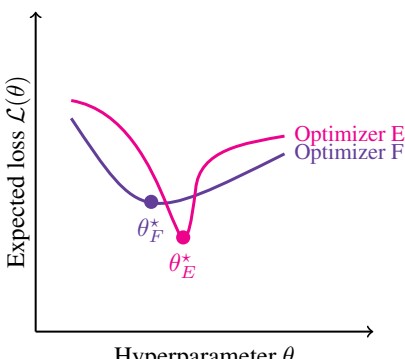

1.a: Illustration. It is important to consider both the absolute performance of optimizers as well as the tuning effort to get to good performances.

1.b: Illustration. While optimizer E can achieve the best performance after careful tuning, optimizer F is likely to provide better performance under a constrained HPO budget.

in most regions of the parameter space F performs much better than E. This makes it easier to find configurations that already perform well. This makes optimizer F an attractive option when we have no prior knowledge of the good parameter settings.

The key-difference between the two interpretations is whether one prefers a 'good enough' performance through fewer hyperparameter configuration searches (in the case of optimizer F), or whether one is willing to spend computational time to get the best possible performance ($\theta_E^\star$ in the case of optimizer E). In this work, we search for the hyperparameter through an HPO like Random Search. Thus, the difference of the two interpretations of tunability lies whether one values results from late stages of the HPO process (i.e. optimizer E is preferable due to better optimum) more than results from early stages of the HPO process (i.e. optimizer F is preferable).

Motivated by these observations, we propose the following metric for tunability.

$\omega$-**tunability's Definition.** *Let $(\boldsymbol{\theta}_t, \mathcal{L}(\boldsymbol{\theta}_t))$ be the incumbents (best performance attained till $t$) of the HPO algorithm at iteration $t$ and $T$ be the hyperparameter tuning budget. For $w_t > 0 \; \forall \; t$ and $\sum_t w_t < \infty$, we define $\omega$-tunability as*

$$\omega\text{-}tunability = \sum_{t=1}^{T} \omega_t \mathcal{L}_t$$

i.e, $\omega$-tunability is a weighted sum of the incumbents $\mathcal{L}(\boldsymbol{\theta}_t)$. In our experiments we use $\sum_t \omega_t = 1$.

By appropriately choosing the weights $\{\omega_t\}$, we can interpolate between our two notions of tunability. In the extreme case where we are only interested in the peak performance of the optimizer, we can set $\omega_T = 1$ and set the other weights to zero. In the opposite extreme case where we are interested in the "one-shot tunability" of the optimizer, we can set $\omega_1 = 1$. In general, we can answer the question of "How well does the optimizer perform with a budget of $K$ iterations?" by setting $\omega_i = \mathbf{1}_{i=K}$.

While the above weighting scheme is intuitive, merely computing the performance after expending HPO budget of $K$ does not consider the performance obtained after the previous $K - 1$ iterations

i.e. we would like to differentiate the cases where a requisite performance is attained by tuning an optimizer for $K$ iterations and another for $K_1$ iterations, where $K_1 \gg K$. Therefore, we employ three additional weighting schemes. By setting $\omega_i \propto (T - i)$, our first one puts more emphasis on the earlier stages of the hyperparameter tuning process. We term this weighting scheme *Cumulative Performance-Early* (*CPE*). In contrast, the second weighting scheme, *Cumulative Performance-Late* (*CPL*) puts more emphasis on late stages of tuning, and thus on obtaining a better performance at a higher tuning cost: $\omega_i \propto i$. As a intermediate of the two, we also report a uniform weighting *Cumulative Performance-Uniform* (*CPU*) : $\omega_i = 1/T$.

## 3    OPTIMIZERS AND THEIR HYPERPARAMETERS

### 3.1    PARAMETERS OF THE OPTIMIZERS

To compare the tunability of adaptive gradient methods to non-adaptive methods, we choose the most commonly used optimizers from both the strata; SGD and SGD with momentum for non-adaptive methods, and Adagrad and Adam for adaptive gradient methods. Since adaptive gradient methods are said to work well with their default hyperparameter values already, we additionally employ a default version of Adam where we only tune the initial learning rate and set the other hyperparameters to the values recommended in the original paper (Kingma & Ba, 2015) (termed AdamLR). Such a scheme has been used by Schneider et al. too. A similar argument can be made for SGD with momentum (termed SGDM): thus we experiment with a fixed momentum value of 0.9 (termed SGDM$^\mathrm{C}$).

In addition to standard parameters in all optimizers, we consider weight decay with SGD too. SGD with weight decay can be considered as an optimizer with two steps where the first step is to scale current weights with the decay value, followed by a normal descent step (Loshchilov & Hutter, 2019). Thus we devise two additional experiments for SGD with weight-decay where we tune weight-decay along with momentum (termed SGDMW), and one where we fix it to $10^{-5}$ (termed SGDM$^\mathrm{C}$W$^\mathrm{C}$) along with the momentum being fixed to 0.9, which is the value for weight decay we found to be consistently better through HPO. The full list of optimizers we consider is provided in Table 4

Manually defining a specific number of epochs can be biased towards one optimizer, as one optimizer may reach good performance in the early epochs of a single HPO iteration, whereas another may reach higher peaks more slowly. In order to alleviate this, it would be possible to add the number of training epochs as an additional hyperparameter to be searched. Since this would incur even higher computational cost, we instead use a validation set performance as stopping criterion. Thus we stop training when the validation loss plateaus for more than 2 epochs or if the number of epochs exceeds the predetermined maximum number as set in DEEPOBS.

### 3.2    CALIBRATION OF HYPERPARAMETER DISTRIBUTIONS

As mentioned previously, we use Random Search for optimizing the hyperparameters, which requires distributions of random variables to sample from. Choosing poor distributions to sample from impacts the performance, and may break requisite properties (e.g. learning rate is non-negative). For some of the parameters listed in Table 2, obvious bounds exist due their mathematical properties, or have been prescribed by the optimizer designers themselves. For example, Kingma & Ba (2015) bound $\beta_1, \beta_2$ to $[0, 1)$ and specify that they are close to 1. In the absence of such prior knowledge, we devise a simple method to determine the priors.

We train each task specified in the DEEPOBS with a large number of hyperparameter samplings and retain the hyperparameters which resulted in performance within $20\%$ of the best performance obtained. For each of the hyperparameters in this set, we fit the distributions in the third column of Table 2 using maximum likelihood estimation. In doing so, we make a simplifying assumption that all the hyperparameters are independent of each other. We argue that these distributions are appropriate; the only condition on learning rate is non-negativity that is inherent to the log-normal distribution, momentum is non-negative with a usual upper bound of 1, $\beta$s in Adam have been prescribed to be less than 1 but close to it, $\epsilon$ is used to avoid divide-by-zero error and thus is a small positive value close to 0. We report the parameters of the distributions obtained after the fitting in Table 2. The calibration procedure's performance is not included in the tunability computation.

Table 2: Optimizers evaluated. For each hyperparameter, we calibrated a 'sampling distribution' to give good results across tasks (Section 3.2). $\mathcal{U}[a, b]$ is the continuous uniform distribution on $[a, b]$. Log-uniform($a$, $b$) is a distribution whose logarithm is $\mathcal{U}[a, b]$. Log-normal($\mu$,$\sigma$) is a distribution whose logarithm is normally distributed with mean $\mu$ and standard deviation $\sigma$.

| Optimizer | Tunable parameters | Sampling distribution |
|---|---|---|
| Stochastic Gradient Descent | Learning rate
Momentum
Weight decay | Log-normal(-2.09, 1.312)
$\mathcal{U}[0, 1]$
Log-uniform(-5, -1) |
| Adagrad | Learning rate | Log-normal(-2.004, 1.20) |
| Adam | Learning rate
$\beta_1, \beta_2$
$\epsilon$ | Log-normal(-2.69, 1.42)
Log-uniform(-5, -1)
Log-uniform(-8, 0) |

Table 3: Models and datasets used. We use the DeepOBS benchmark set (Schneider et al., 2019). Details are provided in Appendix A.

| Architecture | Datasets |
|---|---|
| Convolutional net | FMNIST, CIFAR10/100 |
| Variational autoencoder | FMNIST, MNIST |
| Wide residual network | SVHN |
| Character RNN | Tolstoi's War and Peace |
| Quadratic function | Artificial datatset |
| LSTM | IMDb |

Table 4: Optimizers and tunable parameters. $\gamma$ is learning rate, $\mu$ is momentum, $\lambda$ is the weight decay coefficient.

| Optimizer | Tunable parameters |
|---|---|
| SGD | $\gamma$ ($\mu$=0, $\lambda$=0) |
| SGDM | $\gamma, \mu$ ($\lambda$=0) |
| SGDM$^C$ | $\gamma$ ($\mu$=0.9, $\lambda$=0) |
| SGDM$^C$W$^C$ | $\gamma$ ($\mu$=0.9, $\lambda=10^{-5}$) |
| SGDMW | $\gamma, \mu, \lambda$ |
| Adagrad | $\gamma$ |
| AdamLR | $\gamma$ ($\beta_1$=0.9, $\beta_2$=0.999, $\epsilon=10^{-8}$) |
| Adam | $\gamma, \beta_1, \beta_2, \epsilon$ |

## 4 EXPERIMENTS AND RESULTS

To assess the tunability of optimizers' hyperparameters for the training of deep neural networks, we benchmark using the open-source suite DEEPOBS (Schneider et al., 2019). The architectures and datasets we experiment are given in Table 3. We refer the reader to Schneider et al. (2019) for specific details of the architectures. To obtain a better balance between vision and NLP applications, we added an LSTM network with the task of sentiment classification in the IMDB dataset (Maas et al., 2011), details for which are provided in Appendix A.

### 4.1 HYPERPARAMETERS AND IMPLEMENTATION

The performance of automatic hyperparameter search methods is dependent on its own hyperparameters, which play an important role in the outcome, e.g number of configurations to test. In our experiments, we evaluate 100 configurations with each of the hyperparameter optimization methods. As we use random search, we simulate multiple runs of these 100 configurations through shuffling. This gives us the variance of performance at each step.

### 4.2 ANALYSIS OF TUNABILITY

We analyze the tunability for the various weighting schemes proposed. For the weighting scheme $\omega_i = \mathbf{1}_{i=K}$, for increasing values of $K$, we show the performance as well as its variance in Figure 3. For readability, we show only results for Adam, AdamLR, Adagrad and SGDMW, and the rest are given in Appendix C. It is quite apparent that in most of the tasks, a well tuned SGD with momentum and weight decay is as good as Adam (for large $K$). However, the gap in the performance is quite noticeable when AdamLR outperforms SGD in the VAE tasks and the IMDB task. In the case of image classification problems, SGD variants fare the best, as it has been reported before (Keskar & Socher, 2017). It is interesting to notice that for $K$=4, the decreasing order of variance is nearly always SGDMW, Adam, Adagrad, AdamLR (10 out of 11 cases), even if AdamLR marginally underperforms as it is in the case of Quadratic Deep. Given this formulation, we ask the following

question: given an HPO budget of $K$, what is the best choice for optimizer? We answer this in Appendix D.

For the other weighting schemes, tunability scores are reported in Table 5. We see that there is no one optimizer that is best across three schemes, and tasks presented. A similar trend of SGD doing better than the adaptive gradient methods on image classification tasks is evident. Considering CPE, we observe that AdamLR performs the best in 6 out 9 tasks, where as the other three times SGDM$^C$W$^C$ performs the best. The trend is not very obvious for CPU and CPL. In the case of CPU, AdamLR wins 5 out of 9 tasks, SGDM$^C$W$^C$ wins twice, and SGDM wins once. For CPL, AdamLR wins 4 out of 9, and the SGDM$^C$W$^C$ wins once, and SGDM and SGDM$^C$ win twice each. Summarizing, if peak-performance or even evolving to better performance at a larger hyperparameter search cost, SGD variants are better 5 out 9 times. However, if a good performance is expected in the earlier iterations of hyperparameter search, AdamLR is very competitive. Also, the default parameters of $\beta_1, \beta_2, \epsilon$ of Adam optimizer result in quite good performance, to the point that Adam is rarely the better alternative over AdamLR. A known exception is training Inception networks (Abadi et al., 2015), where $\epsilon$ is recommended to be set to 0.1.

For some of the cases, the tunablities reported are very similar for the AdamLR and SGD variants. However, tuning Adam is very different from tuning SGD from a wall-clock time measurement. For example, we find that for the case of CIFAR-10, AdamLR requires on average 39% fewer epochs to complete training than SGDM$^C$ (the top perfomer); thus being that much faster than SGDM$^C$ in wall-clock time. It can be argued that a more practical form of hyperparameter tuning budget is wall-clock time i.e. if a wall-clock time budget of $K$ minutes is given, how do our findings vary? In short, we find similar trends as we noticed before. We provide the details in Appendix E.

### 4.3 SUMMARIZING ACROSS DATASETS

To get a better understanding of an optimizer's aggregate tunability across datasets compared to the rest, we compute summary statistics for an optimizer $o$'s performance after $k$ iterations in the following way:

$$S(o, k) = \frac{1}{|\mathcal{P}|} \sum_{p \in \mathcal{P}} \frac{o(k, p)}{\max_{o' \in \mathcal{O}} o'(k, p)},$$

where $o(k, p)$ denotes the performance of optimizer $o \in \mathcal{O}$ on test problem $p \in \mathcal{P}$ after $k$ iterations of the HPO process (i.e. $\omega$-tunability with $\omega_i = \mathbf{1}_{i=k}$). In other words, we compute the average relative performance of an optimizer to the best performance of any optimizer over all tasks.

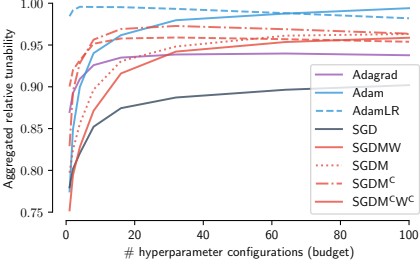

Figure 2: Aggregated relative tunability of each optimizer across datasets

The results are in Figure 2 and show that AdamLR performs very close to the best optimizer throughout the HPO process and is the best till about the 60th iteration. In early stages of HPO, the SGD variants perform 10–20% worse than Adam, but improve as the HPO progresses.

## 5 RELATED WORK

There exist few works that have tried to define and investigate tunability formally. Assessing the impact of hyperparameter tuning for decision tree models, Mantovani et al. (2018) count the number of times the tuned hyperparameter values are (statistically significantly) better than the default values. Probst et al. (2019) define tunability of an ML algorithm as the performance difference between a reference configuration (e.g., the default hyperparameters of the algorithm) and the best possible configuration on each dataset. This metric is comparable across ML algorithms, but it disregards entirely the absolute performance of ML algorithms. Schneider et al. (2019) recently released a benchmark for optimizers that evaluates their peak performance and speed. Tunability is assessed as the sensitivity of the performance to changes of the learning rate. In all three aforementioned studies, the definitions of tunability would fail to identify the superiority of optimizer A over optimizer B in Figure 1.a.

The study by Wilson et al. (2017) finds SGD-based methods as easy to tune as adaptive gradient methods. However, their study lacks a clear definition of tunability and tunes the algorithms on

Table 5: Performance of various experiments.

| Optimizer | CPE | CPU | CPL |
|---|---|---|---|
| Adagrad | **91.3** | 91.4 | 91.6 |
| Adam | **91.3** | 91.5 | 91.8 |
| AdamLR | **91.3** | **91.6** | **91.9** |
| SGD | 90.4 | 90.8 | 91.2 |
| SGDM | 90.5 | 90.9 | 91.3 |
| $\text{SGDM}^C$ | 90.7 | 90.9 | 91.1 |
| $\text{SGDM}^C\text{W}^C$ | 90.7 | 90.9 | 91.1 |
| SGDMW | 90.4 | 90.8 | 91.3 |

5.a: FMNIST 2C2D. Higher is better

| Optimizer | CPE | CPU | CPL |
|---|---|---|---|
| Adagrad | 76.4 | 77.1 | 77.9 |
| Adam | 77.2 | 78.4 | 79.5 |
| AdamLR | 78.8 | 79.4 | 80.0 |
| SGD | 77.0 | 77.8 | 78.6 |
| SGDM | 77.8 | 78.6 | 79.5 |
| $\text{SGDM}^C$ | 78.6 | 79.4 | 80.1 |
| $\text{SGDM}^C\text{W}^C$ | **81.1** | **81.6** | **82.0** |
| SGDMW | 79.7 | 80.4 | 81.2 |

5.b: CIFAR 10. Higher is better

| Optimizer | CPE | CPU | CPL |
|---|---|---|---|
| Adagrad | 30.4 | 31.8 | 33.1 |
| Adam | 39.4 | 42.2 | 45.1 |
| AdamLR | **42.2** | 43.0 | 43.8 |
| SGD | 31.8 | 34.2 | 36.6 |
| SGDM | 40.6 | **43.3** | **46.0** |
| $\text{SGDM}^C$ | 42.1 | **43.3** | 44.5 |
| $\text{SGDM}^C\text{W}^C$ | 39.2 | 40.3 | 41.5 |
| SGDMW | 33.5 | 37.2 | 41.0 |

5.c: CIFAR 100. Higher the better

| Optimizer | CPE | CPU | CPL |
|---|---|---|---|
| Adagrad | 84.3 | 84.8 | 85.3 |
| Adam | 83.6 | 84.5 | 85.5 |
| AdamLR | **85.8** | **86.0** | **86.3** |
| SGD | 68.1 | 69.3 | 70.5 |
| SGDM | 74.3 | 75.9 | 77.5 |
| $\text{SGDM}^C$ | 79.3 | 80.1 | 81.0 |
| $\text{SGDM}^C\text{W}^C$ | 78.8 | 79.4 | 80.0 |
| SGDMW | 75.7 | 77.1 | 78.6 |

5.d: IMDB. Higher is better

| Optimizer | CPE | CPU | CPL |
|---|---|---|---|
| Adagrad | 94.8 | 94.9 | 95.0 |
| Adam | 94.5 | 94.8 | 95.2 |
| AdamLR | 95.1 | 95.3 | 95.4 |
| SGD | 94.6 | 94.9 | 95.2 |
| SGDM | 94.8 | 95.2 | **95.6** |
| $\text{SGDM}^C$ | 94.9 | 95.1 | 95.3 |
| $\text{SGDM}^C\text{W}^C$ | **95.2** | **95.4** | 95.5 |
| SGDMW | 95.0 | 95.2 | 95.3 |

5.e: WRN-16(4). Higher is better

| Optimizer | CPE | CPU | CPL |
|---|---|---|---|
| Adagrad | 55.6 | 56.2 | 56.7 |
| Adam | 54.4 | 55.7 | 57.0 |
| AdamLR | **56.9** | **57.2** | 57.5 |
| SGD | 40.3 | 42.5 | 44.6 |
| SGDM | 51.4 | 54.0 | 56.5 |
| $\text{SGDM}^C$ | 55.6 | 57.0 | **58.3** |
| $\text{SGDM}^C\text{W}^C$ | 54.2 | 55.6 | 57.0 |
| SGDMW | 45.1 | 48.2 | 51.2 |

5.f: Char-RNN. Higher is better

| Optimizer | CPE | CPU | CPL |
|---|---|---|---|
| Adagrad | 25.5 | 24.7 | 24.0 |
| Adam | 26.0 | 24.8 | 23.7 |
| AdamLR | **24.6** | **24.0** | **23.5** |
| SGD | 26.2 | 25.5 | 24.8 |
| SGDM | 26.2 | 25.4 | 24.6 |
| $\text{SGDM}^C$ | 28.6 | 26.7 | 24.9 |
| $\text{SGDM}^C\text{W}^C$ | 27.9 | 26.4 | 24.8 |
| SGDMW | 26.5 | 25.7 | 24.8 |

5.g: FMNIST-VAE. Lower is better.

| Optimizer | CPE | CPU | CPL |
|---|---|---|---|
| Adagrad | 30.3 | 29.4 | 28.4 |
| Adam | 33.2 | 31.2 | 29.1 |
| AdamLR | **29.2** | **28.6** | **27.9** |
| SGD | 53.3 | 53.1 | 52.9 |
| SGDM | 36.0 | 32.9 | 29.9 |
| $\text{SGDM}^C$ | 54.1 | 53.5 | 53.0 |
| $\text{SGDM}^C\text{W}^C$ | 54.0 | 53.5 | 53.0 |
| SGDMW | 34.6 | 32.2 | 29.8 |

5.h: MNIST-VAE. Lower is better

| Optimizer | CPE | CPU | CPL |
|---|---|---|---|
| Adagrad | 91.5 | 89.6 | 87.6 |
| Adam | 94.8 | 92.1 | 89.4 |
| AdamLR | 91.2 | 89.5 | 87.7 |
| SGD | 90.5 | 89.6 | 88.7 |
| SGDM | 89.5 | 88.7 | 87.9 |
| $\text{SGDM}^C$ | 89.6 | 88.6 | **87.5** |
| $\text{SGDM}^C\text{W}^C$ | **88.6** | **88.4** | 88.1 |
| SGDMW | 89.3 | 88.9 | 88.4 |

5.i: Quadratic deep. Lower is better

manually selected, dataset dependent grid values. The study by Shah et al. (2018) applies a similar methodology and comes to similar conclusions regarding tunability. Since both studies only consider the best parameter configuration, their approach would be unable to identify the better optimizer among B and D in Figure 1.a. In contrast, the methodology in our study is able to distinguish all the cases depicted in Figure 1.a.

In a concurrent study, Choi et al. (2019) show that there exist a hierarchy among optimizers that such some can be viewed as specific cases of others e.g. SGDM is shown to be a special case of Adam as its $\epsilon \to \infty$, and thus Adam should never underperform SGDM with appropriate hyperparameter search). Like in our study, they suggest that the performance comparison of optimizers strongly depends on the hyperparameter tuning protocol. They also argue that the search space needs to be chosen optimizer specific. However, their focus is on the best possible performance achievable by an optimizer and does not take into account the tuning process. Moreover, while the authors claim their search protocol to be relevant for practitioners, the search spaces are manually chosen *per dataset*, constituting a significant difference to the AutoML scenario considered in our paper.

Tunability is related to measuring hyperparameter importance (Hutter et al., 2013), where van Rijn & Hutter (2018) have recently shown that learning the priors for hyperparameter distributions can yield to better HPO performance, akin to the calibration phase in our study.

There has been recent interest in building optimizers termed the APROX family (Asi & Duchi, 2019a;b) that are provably robust to hyperparameter choices. Asi & Duchi experimentally find that, training a Residual network (He et al., 2016) on CIFAR-10, SGD converges only for a small range of initial learning rate choices, whereas Adam exhibits better robustness to learning rate choices. This is inline with our findings of tunability.

## 6 CONCLUSION

Our work proposes a new notion of *tunability* for optimizers that takes into account the tuning efforts of an HPO. The results of our experiments support the hypothesis that adaptive gradient methods are easier to tune than non-adaptive methods: In a setting with low budget for hyperparameter tuning, tuning only Adam optimizer's learning rate is likely to be a very good choice; it doesn't guarantee the best possible performance, but it is evidently the easiest to find well-performing hyperparameter configurations for. While SGD yields the best performance in some cases, its best configuration is tedious to find, and Adam often performs close to it. We, thus, state that the substantial value

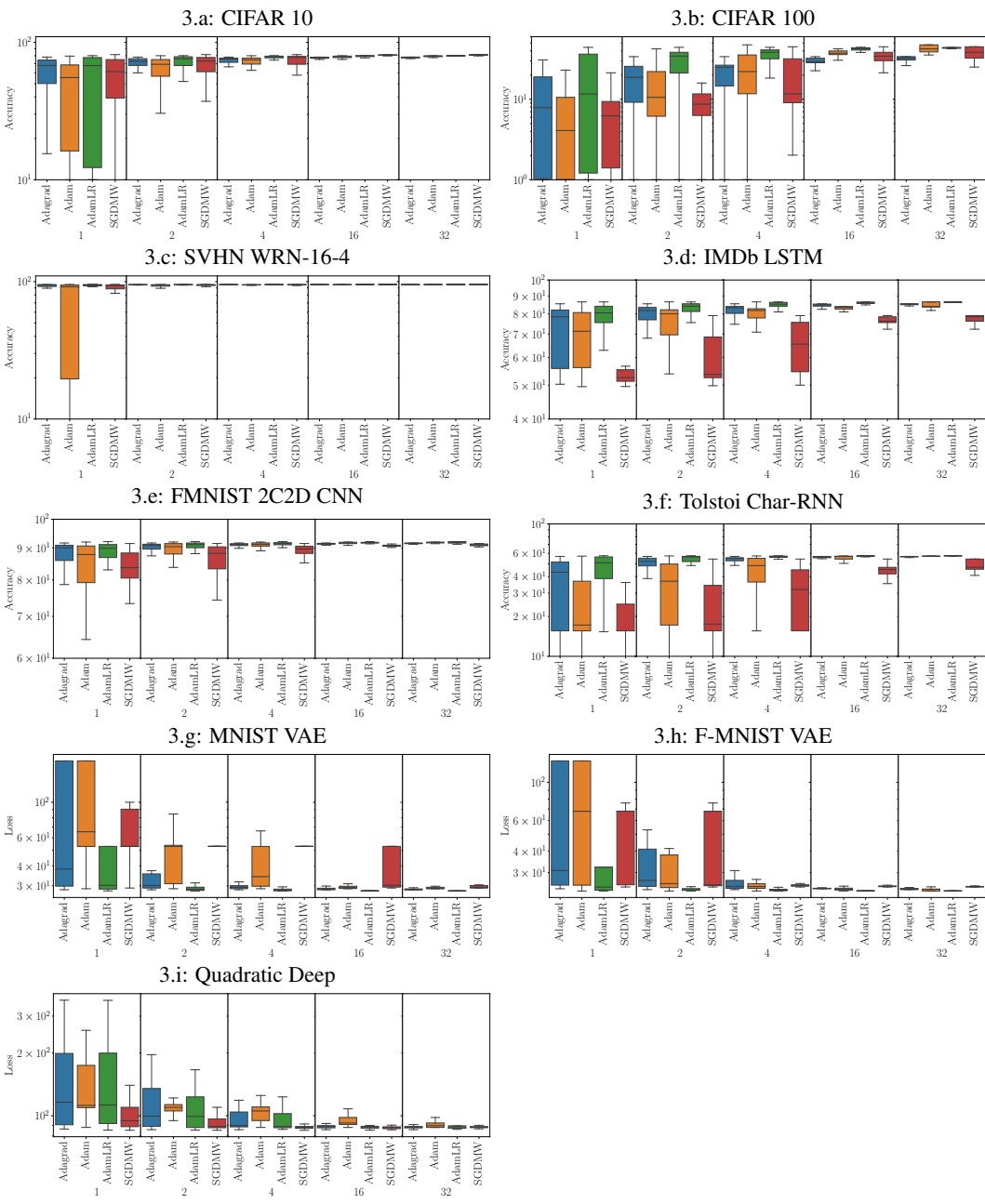

Figure 3: $\omega$-tunability with $\omega_i = \mathbf{1}_{i=K}$ for various experiments. We plot the on the x-axis the number of the hyperparameter configuration searches, on the y-axis the appropriate performance on a log scale. Figures a-f: higher is better and g-i: lower is better.

of the adaptive gradient methods, specifically Adam, is its amenability to hyperparameter search. This is in contrast to the findings of Wilson et al. (2017) who observe no advantage in tunabilty for adaptive gradient methods, and thus deem them to be of 'marginal value'. Unlike them, we base our experiments on a standard hyperparameter optimization method that allows for an arguably fairer comparison.

Our study is certainly not exhaustive: We do not study the effect of the inclusion of a learning rate schedule, or using a different HPO algorithm on the results. However, their inclusion would result in a large increase the number of experiments, and constitutes our future work.

We hope that this paper encourages other researchers to conduct future studies on the performance of optimizers from a more holistic perspective, where the cost of the hyperparameter search is included.

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

## A    ARCHITECTURES OF THE MODELS USED IN EXPERIMENTS

Along with the architectures examined by Schneider et al. (2019), we experiment with an additional network and dataset. We included an additional network into our experimental setup, as DEEPOBS does not contain an word level LSTM model. Our model uses a 32-dimensional word embedding table and a single layer LSTM with memory cell size 128, the exact architecture is given in Table 6. We experiment with the IMDB sentiment classification dataset (Maas et al., 2011). The dataset contains $50,000$ movie reviews collected from movie rating website IMDB. The training set has $25,000$ reviews, each labeled as positive or negative. The rest $25,000$ form the test set. We split $20\%$ of the training set to use as the development set. We refer the readers to DEEPOBS (Schneider et al., 2019) for the exact details of the other architectures used in this work.

Table 6: Architecture of the LSTM network used for IMDb experiments

| Layer name | Description |
|---|---|
| Emb | $\begin{bmatrix} \text{Embedding Layer} \\ \text{Vocabulary of 10000} \\ \text{Embedding dimension: 32} \end{bmatrix}$ |
| LSTM_1 | $\begin{bmatrix} \text{LSTM} \\ \text{Input size: 32} \\ \text{Hidden dimension: 128} \end{bmatrix}$ |
| FC Layer | $\text{Linear}(128 \rightarrow 2)$ |
| Classifier | $\text{Softmax}(2)$ |

## B    $\alpha$- TUNABILITY

We provide additional methods to analyze tunability here. Let $p(t)$ denote the best performance observed after using budget $t$ of hyperparameter optimization algorithm. We call an optimizer $\alpha$-tunable ($\alpha \in [0,1]$) at $t$ if $p(t) \geq \alpha \cdot p(T)$. Thus $\alpha-$tunability is the ratio of number of times the neural network needs to be retrained with optimizer's hyperparameters being provided by an automatic method, to the total budget $T$ (maximum number of configurations tested).

For each optimizer, we define its $\alpha$-*tunability* $\zeta(\alpha) = \frac{t}{T}$ for $\alpha \in \{0.9, 0.95, 0.99\}$. This metric provides an intuitive and simple quantification of how easy it is to tune an optimizer to reach requisite performance. We extend $\alpha$–tunability to indicate the sharpness of the minima by computing the difference $\Delta = \zeta(\alpha_1) - \zeta(\alpha_2)$ where $\alpha_1 > \alpha_2$ and term it *Sharpness*. In our experiments, we choose $\alpha_1 = 0.99$ and $\alpha_2 = 0.9$. Sharpness($\Delta$) is the relative time taken by the HPO to improve from $\alpha_2$ to $\alpha_1$ and thus quantifies the flatness of the minima in the space of hyperparameters. If the minima is sharper, then we expect random-search also takes a longer time to find it, thus the time required to go from $\alpha_1$ and $\alpha_2$ is higher. We provide Sharpness for our optimizers in table:7.

| | MNIST VAE | FMNIST 2C2D | CIFAR 100 | CIFAR 10 | SVHN WRN | IMDB LSTM | FMNIST VAE | Quadratic Deep | Char RNN |
|---|---|---|---|---|---|---|---|---|---|
| **Adagrad** | 92.0 | 99.0 | 63.0 | 95.0 | **10.0** | 97.0 | 91.0 | 66.0 | 95.0 |
| **Adam** | 80.0 | 98.0 | 40.0 | **91.0** | 71.0 | 93.0 | 90.0 | 75.0 | 87.0 |
| **Adam LR** | 83.0 | 96.0 | 71.0 | 94.0 | 98.0 | 70.0 | 95.0 | 93.0 | 97.0 |
| **SGD** | 98.0 | 95.0 | 35.0 | 93.0 | 98.0 | 81.0 | 94.0 | **6.0** | 47.0 |
| **SGDM** | **58.0** | 97.0 | 50.0 | **91.0** | 59.0 | 72.0 | 90.0 | 30.0 | 59.0 |
| **SGDM$^C$** | 95.0 | 83.0 | 82.0 | 94.0 | 52.0 | 88.0 | **75.0** | 95.0 | 87.0 |
| **SGDM$^C$W$^C$** | 94.0 | **45.0** | 78.0 | 95.0 | 98.0 | **68.0** | 88.0 | 16.0 | 85.0 |
| **SGDMW** | 65.0 | 97.0 | **23.0** | 94.0 | 36.0 | 86.0 | 91.0 | 14.0 | **32.0** |

Table 7: Sharpness for various optimizers examined.

The above definition is not without faults. An optimizer's $\alpha$-tunability depends only on how fast it can get close to its *own best performance*, a pitfall it shares with Probst et al. (2019). That is, an optimizer that peaks at the performance of a random classifier may be considered well-tunable because it reaches its peak performance in the first iteration. It is apparent from tables 7 and 4 that

the top performance does not imply lower sharpness. Take the case of IMDB Bi-LSTM, the lowest sharpness is for SGDM$^C$W$^C$, while the best performance is attained by AdamLR, implying that SGDM$^C$W$^C$ settled to a minima faster which isn't necessarily better than the one AdamLR found. In other terms, the flatness of the minima does not indicate how deep it is.

## C   PERFORMANCE ANALYSIS

We show the full performance plots of all variants of SGD experimented with, in figures 5, 6, 7.

## D   HOW LIKELY ARE WE TO FIND GOOD CONFIGURATIONS?

A natural question that arises is: given a budget $K$, what is the best optimizer one can pick? In other words, for a given budget what is probability of each optimizer finding the best configuration? We answer this with a simple procedure. We repeat the runs of HPO for a budget $K$, and collect the optimizer that gave the best result in each of those runs. Using the classical definition of probability, we compute the required quantity. We plot the computed probability in Figure 8. It is very evident for nearly all budgets, AdamLR is always the best option for 4 of the problems. SGD variants emerge to be better options for CIFAR-100 and Char-RNN at later stages of HPO. For some of the problems like VAEs, LSTM, it is very obvious that AdamLR is nearly always the best choice. Thus further strengthens our hypothesis that adaptive gradient methods are more tunable, especially in constrained HPO budget scenarios.

## E   TUNABILITY BY COMPUTATION BUDGET

In our experiments so far, we defined tunability in terms of number of hyperparameter configurations. However, due to varying convergence speeds, different optimizers require varying amounts of time/epochs per configuration. We choose to work with number of epochs per configuration, as it is a hardware agnostic measure, but still indicates the relative time required. To incorporate epochs into our definition of tunability, we conduct the following analysis: For each dataset, we consider minimum total epochs for running all 100 trials across all optimizers, and consider it as the (virtual) maximum epoch budget $e_{\max}$, i.e., we disregard all trials after this point. We divide this maximum into $K = 100$ intervals $I_i = \frac{e_{\max} \cdot i}{K}, 1 \leq i \leq K$. We modify the definition of $\omega$-tunability from Section 2.2 as follows:

$\boldsymbol{\omega}^{epoch}$**-tunability's Definition.** *Let $(\boldsymbol{\theta}_t, \mathcal{L}(\boldsymbol{\theta}_t))$ be the incumbents (best performance attained till t) of the HPO algorithm at iteration t, $e_t$ be the total number of epochs required until iteration t has finished. We define $\tilde{\mathcal{L}}_i = \max_{e_t \leq I_i} \mathcal{L}_t$ as the maximum performance among all configurations that have finished before interval $I_t$, where $\mathcal{L}_0 = e_0 = 0$. For $w_i > 0 \, \forall \, t$ and $\sum_t w_i < \infty$, we define $\omega^{epoch}$-tunability as*

$$\boldsymbol{\omega}^{epoch}\text{-tunability} = \sum_{i=1}^{K} \omega_i \tilde{\mathcal{L}}_i$$

Please note that due to the case where no trial has finished before the first interval has concluded, we assign a performance of 0 for that task to that interval (bin). The above definition does not lend itself to VAE tasks, as an appropriate value there is $\infty$. We, therefore, report the results for all classification problems in Table 8, where we use the same weighting schemes (CPE, CPU, and CPL) as before. To better understand the results, we also report the average number of epochs each configuration takes on average. Comparing the relative performances of optimizers to the results from Table 5, we can observe that considering $\boldsymbol{\omega}^{epoch}$ yields to the same conclusions as before, and even amplifies Adam's strengths: The performance gap to SGD variants widens in the cases where Adam(LR) was already better (FMNIST, IMDB, Char-RNN), and narrows considerably in the cases where an SGD variant was previously better (CIFAR-10, CIFAR-100, WRN-16). On CIFAR-100 and WRN-16, this even results in Adam outperforming the SGD variants slightly. Considering the average number of epochs, the results can easily be explained by the fact that the adaptive gradient methods tend to take less time to converge.

Table 8: $\omega^{epoch}$-tunability performance of optimizers on the classification tasks for CPE, CPU, and CPL weighting schemes. We additionally provide the average number of epochs required by each optimizer for a single configuration.

| Optimizer | CPE | CPU | CPL | Epochs |
|-----------|-----|-----|-----|--------|
| Adagrad | 89.2 | 90.3 | 91.4 | 33.0 |
| Adam | 90.2 | 91.0 | 91.8 | 24.1 |
| AdamLR | **90.4** | **91.1** | **91.9** | **19.5** |
| SGD | 87.4 | 89.2 | 90.9 | 34.5 |
| SGDM | 88.1 | 89.6 | 91.1 | 33.6 |
| SGDMC | 88.6 | 89.8 | 91.0 | 28.2 |
| SGDMCWC | 89.1 | 90.1 | 91.1 | 27.4 |
| SGDMW | 88.0 | 89.5 | 91.0 | 32.4 |

8.a: FMNIST 2C2D. Higher is better

| Optimizer | CPE | CPU | CPL | Epochs |
|-----------|-----|-----|-----|--------|
| Adagrad | 74.2 | 75.9 | 77.5 | 36.9 |
| Adam | 75.6 | 77.5 | 79.3 | 27.7 |
| AdamLR | **78.2** | 79.1 | 80.0 | **24.9** |
| SGD | 73.4 | 75.6 | 77.9 | 49.6 |
| SGDM | 74.3 | 76.5 | 78.8 | 46.9 |
| SGDMC | 75.6 | 77.6 | 79.6 | 44.4 |
| SGDMCWC | **78.2** | **79.9** | **81.7** | 47.6 |
| SGDMW | 75.5 | 78.1 | 80.6 | 52.4 |

8.b: CIFAR 10. Higher is better

| Optimizer | CPE | CPU | CPL | Epochs |
|-----------|-----|-----|-----|--------|
| Adagrad | 30.1 | 31.6 | 33.1 | **23.7** |
| Adam | 36.0 | 39.4 | 42.8 | 36.8 |
| AdamLR | **41.8** | **42.8** | **43.8** | 24.6 |
| SGD | 23.3 | 26.6 | 29.9 | 91.8 |
| SGDM | 30.5 | 34.7 | 39.0 | 80.6 |
| SGDMC | 36.9 | 39.8 | 42.7 | 68.7 |
| SGDMCWC | 34.1 | 36.8 | 39.4 | 71.7 |
| SGDMW | 21.5 | 25.6 | 29.7 | 102.3 |

8.c: CIFAR 100. Higher the better

| Optimizer | CPE | CPU | CPL | Epochs |
|-----------|-----|-----|-----|--------|
| Adagrad | 83.1 | 84.1 | 85.2 | 34.4 |
| Adam | 82.1 | 83.7 | 85.2 | 31.2 |
| AdamLR | **84.8** | **85.5** | **86.2** | **28.6** |
| SGD | 65.8 | 67.7 | 69.6 | 42.2 |
| SGDM | 72.2 | 74.5 | 76.9 | 37.2 |
| SGDMC | 77.9 | 79.3 | 80.8 | 32.3 |
| SGDMCWC | 71.3 | 75.0 | 78.6 | 109.7 |
| SGDMW | 64.6 | 69.6 | 74.6 | 119.0 |

8.d: IMDB. Higher is better

| Optimizer | CPE | CPU | CPL | Epochs |
|-----------|-----|-----|-----|--------|
| Adagrad | 93.8 | 94.4 | 95.0 | 19.7 |
| Adam | 92.0 | 93.5 | 95.1 | 24.3 |
| AdamLR | **94.5** | **95.0** | 95.4 | **17.2** |
| SGD | 92.6 | 93.8 | 95.1 | 29.9 |
| SGDM | 91.9 | 93.6 | 95.3 | 29.2 |
| SGDMC | 93.3 | 94.2 | 95.2 | 26.0 |
| SGDMCWC | 92.5 | 94.0 | **95.4** | 31.8 |
| SGDMW | 91.6 | 93.4 | 95.3 | 33.5 |

8.e: WRN-16(4). Higher is better

| Optimizer | CPE | CPU | CPL | Epochs |
|-----------|-----|-----|-----|--------|
| Adagrad | 54.5 | 55.5 | 56.6 | 166.6 |
| Adam | 53.7 | 55.3 | 57.0 | **131.0** |
| AdamLR | **55.9** | **56.7** | 57.4 | 170.9 |
| SGD | 38.1 | 40.8 | 43.4 | 183.2 |
| SGDM | 48.5 | 51.8 | 55.2 | 188.3 |
| SGDMC | 53.3 | 55.6 | **58.0** | 194.5 |
| SGDMCWC | 51.7 | 54.2 | 56.6 | 197.3 |
| SGDMW | 42.2 | 45.6 | 49.1 | 184.0 |

8.f: Char-RNN. Higher is better

Figure 4: Performance analysis of various experiments. We plot the on the x-axis the number of the hyperparameter configuration searches, on the y-axis the appropriate performance.

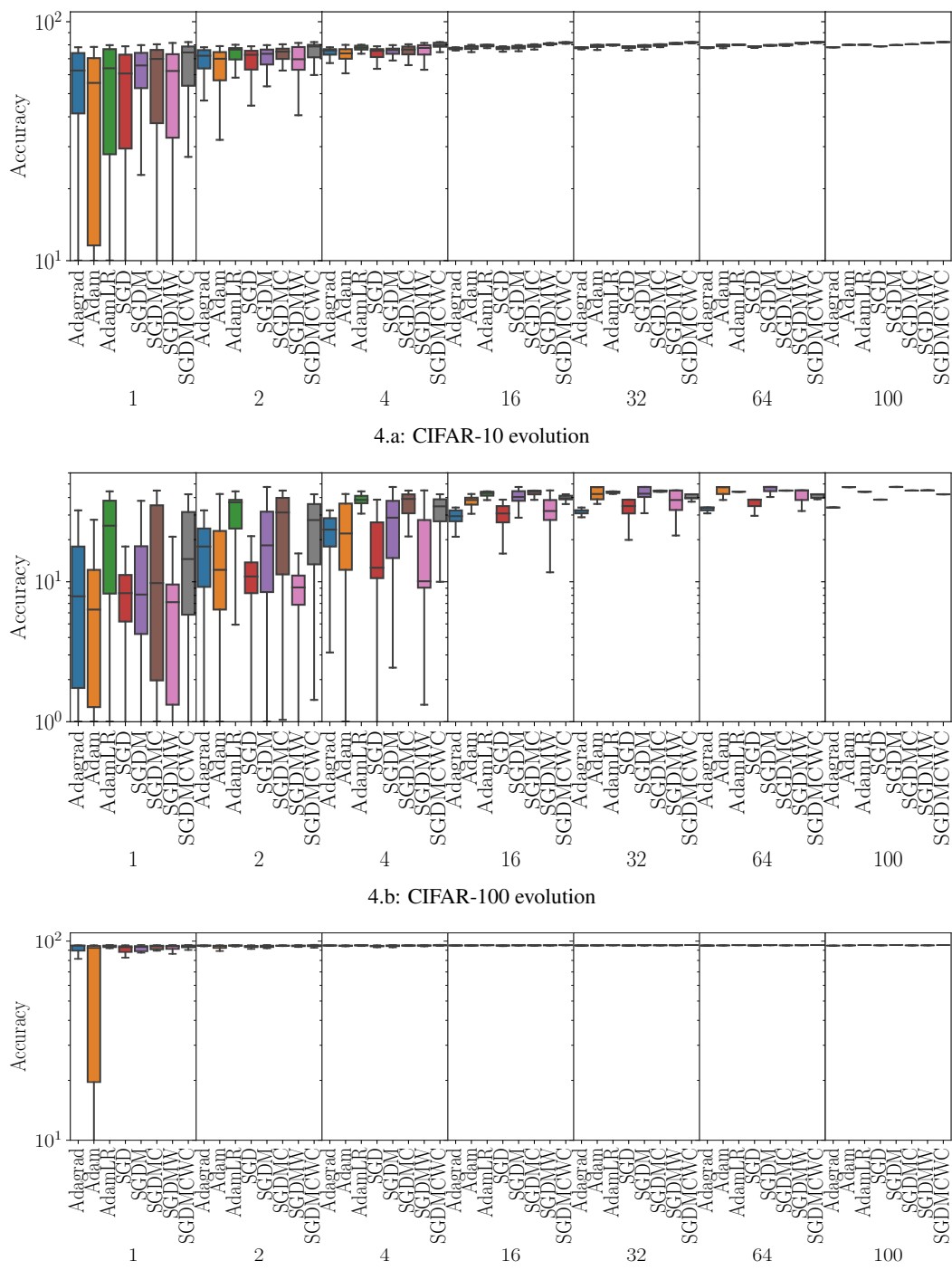

4.a: CIFAR-10 evolution

4.b: CIFAR-100 evolution

4.c: WRN evolution

Figure 5: Performance analysis of various experiments. We plot the on the x-axis the number of the hyperparameter configuration searches, on the y-axis the appropriate performance on a log scale

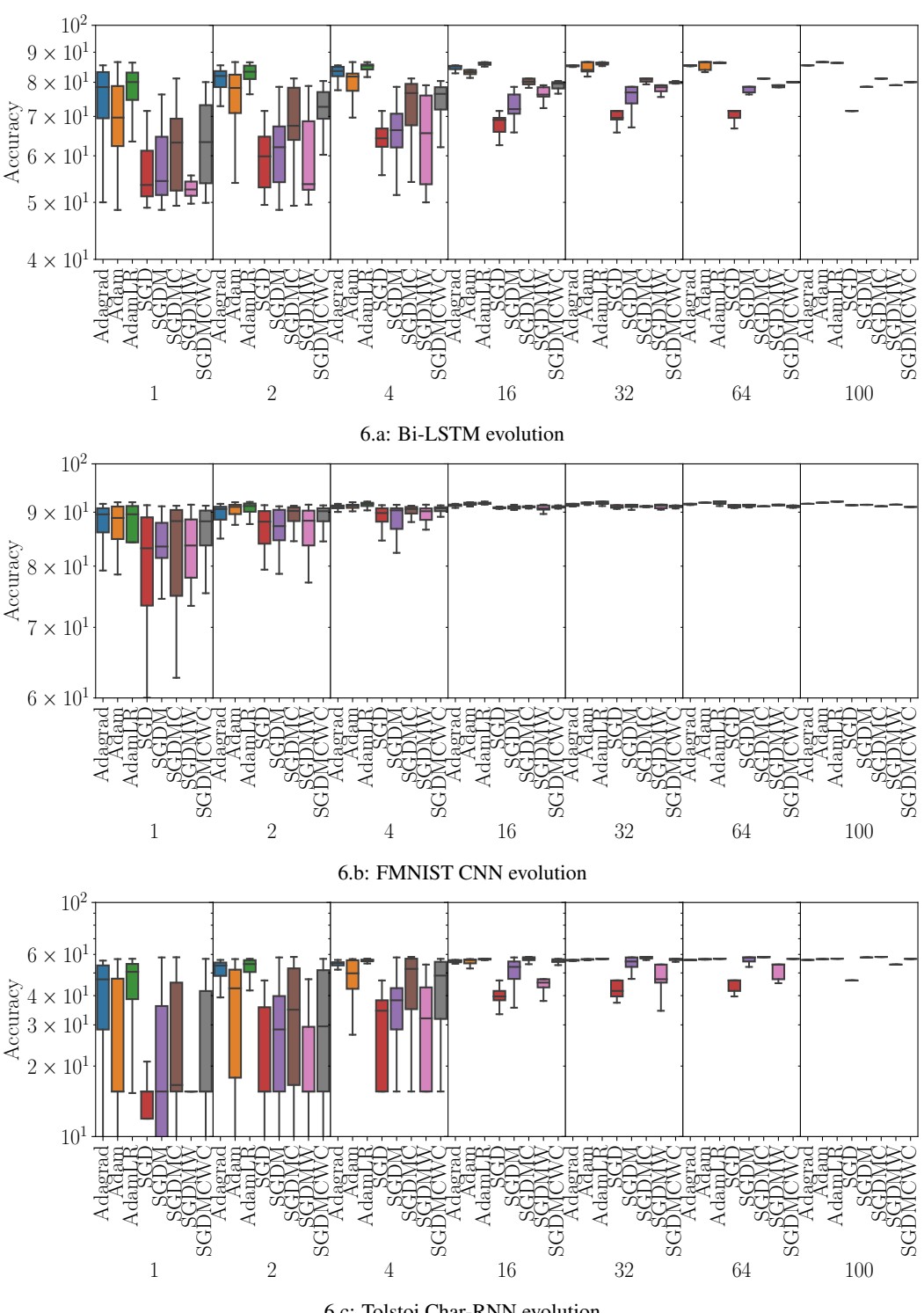

6.a: Bi-LSTM evolution

6.b: FMNIST CNN evolution

6.c: Tolstoi Char-RNN evolution

Figure 6: Performance analysis of various experiments. We plot the on the x-axis the number of the hyperparameter configuration searches, on the y-axis the appropriate performance on a log scale

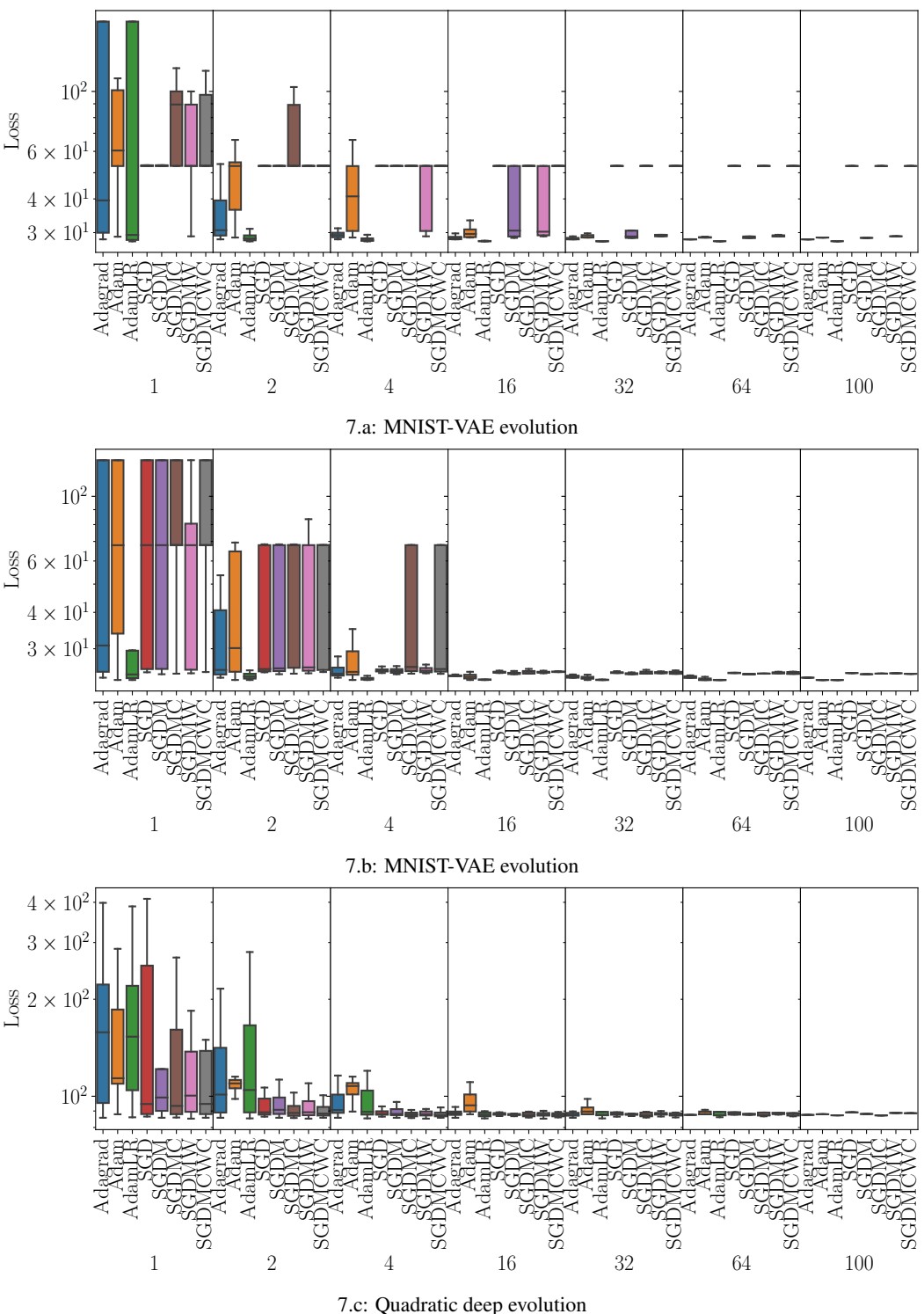

7.a: MNIST-VAE evolution

7.b: MNIST-VAE evolution

7.c: Quadratic deep evolution

Figure 7: Performance analysis of various experiments. We plot the on the x-axis the number of the hyperparameter configuration searches, on the y-axis the appropriate performance on a log scale

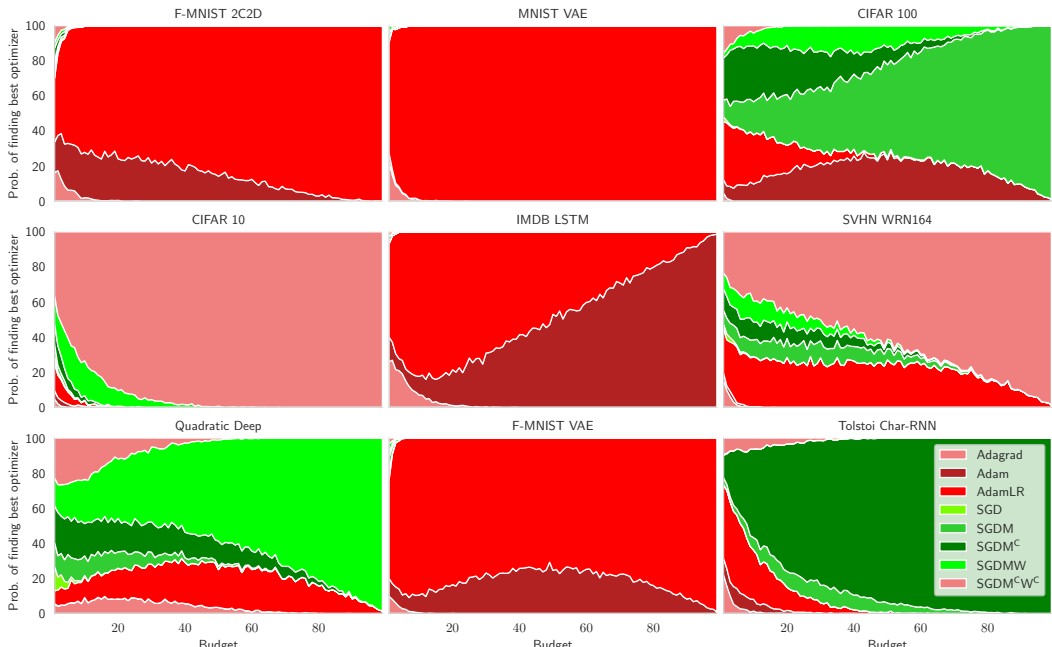

Figure 8: Which optimizer for which budget? Given a tuning budget $K$ ($x$-axis), the stacked area plots above show how likely each optimizer (colored bands) is to yield the best result after $K$ steps of hyperparameter optimization. For example, for the IMDB LSTM problem, for a small budget, 'AdamLR' is the best choice (with $\sim 0.8$ probability), whereas for a larger search budget $> 50$, tuning the additional parameters of 'Adam' is likely to pay off.

