# OpenReview forum: "On the Tunability of Optimizers in Deep Learning"
_ICLR.cc/2020/Conference — Reject_

### Official Review · AnonReviewer1 · 2019-10-23
**Official Blind Review #1**

**Rating:** 3

**Review:**

The main contributions of the submission are:

1. A comprehensive empirical comparison of deep learning optimizers, with their performance compared under different amount of hyper-parameter tuning (they perform hyper-parameter tuning using random search).
2. The introduction of a novel metric that tries to capture the "tunability" of an optimizer. This metric attempts to trade off the performance of an optimizer when tuned only with a small number of hyper-parameter trials, and its performance when carefully tuned. The metric is defined as a weighted average of the performance after tuning with i random trials, with i that goes from 1 to K. The weights of this weighted average and K are "hyper-parameters" of the metric itself. They use K=100 and suggest 3 possible choices of weights.

The paper appears to treat 2. as the main contribution. However, I do not think the metric they introduce is good enough to be recommended in future work, when comparing tunability of optimizers (or other algorithms with hyperparameters). The reason is that simpler methods provide just as much information, and do not rely on the need of interpreting the choice of the weights and K. This point is proven in the paper itself, where for example Figure 2 provides a more concrete and easier to interpret information than the tunability metric, similar graphs could be easily provided per dataset. Similarly, figure 3 as well as figures 5-7 and 8 in the appendix provide very good information about the tunability of the various optimizers without using the introduced metrics. Information similar (although not identical) to that summarized in table 5 could be captured by substituting the 3 metrics with the best performance after tuning for 4, 16 and 64 iterations respectively (just as examples).

A stronger contribution is 1., which however is somewhat incremental compared to similar comparisons made in the past. Comparisons which, while mentioned, should perhaps have been discussed and compared more in detail in this work. Overall, I do not feel the comparisons dramatically change the qualitative understanding the field has of the different optimizers and their tunability.

They also suggest that when the tuning budget is low, using Adam but tuning only the learning rate is beneficial, which could be a valuable and practical suggestion.

I enjoyed reading the submission, which is very clearly written, but due to the relatively limited value of the contributions, and excessive focus on the tunability metric which I do not feel is giustified, I slightly lean against acceptance here at ICLR. I do think, however, that it would make a great submission to a smaller venue or workshop.

Other comments/notes:
* One aspects that is mostly left out of the discussion (except from one side comment) is the wallclock time, as some optimizers might be on average quicker to train (for example due to quicker convergence), this can easily lead it to be quicker to tune even though it requires a higher budget of trials. I think it would be worth discussing this more.
* minor: in figure 8 in the appendix, the results after 100 iterations is, as far as I understand, over a single replication, so is not particularly reliable (and will always be 100% of a single optimizer)
* similarly to the above, if the configurations are always sampled from the same 100, confidence intervals in the graphs become less reliable as the budget increases.

**Experience Assessment:**

I have read many papers in this area.

**Review Assessment: Checking Correctness Of Derivations And Theory:**

I carefully checked the derivations and theory.

**Review Assessment: Checking Correctness Of Experiments:**

I assessed the sensibility of the experiments.

**Review Assessment: Thoroughness In Paper Reading:**

I read the paper thoroughly.

---

> ### Author Response · Authors · 2019-11-12
> **Our main contribution is a fair evaluation protocol for optimizer benchmarking**
>
> Thank you for your review of our work.
>
> The following are your concerns of our work:
> a. Limited contribution to the qualitative understanding of the optimizers
>     a.i. Informativeness of the proposed w-tunability metric
> b. Using wall-clock time instead of number of HPO oracle calls
>
> We address these concerns one-by-one.
>
> a. Limited contribution to the qualitative understanding of the optimizers:
> We consider the three main contributions of our work to be 1) a systematic evaluation protocol of optimizers, with off-the-shelf HPO to account for the cost of tuning of hyperparameters. This is missing in existing papers, which consider best attained performance alone. The importance of a proper hyperparameter search protocol is emphasized by Choi et al., 2019 (published after our submission and under review at ICLR). 2) a “w-tunability” measure of the cost of hyperparameter optimization, and 3) under the experiments considered we find that Adam (with default beta and epsilon values) is the most tunable.
>
> a.i. Informativeness of the tunability metric:
> We propose w-tunability as a metric to incorporate the HPO tuning too in reporting the performance of an optimizer, and compute it as a linear combination of the incumbents of the HPO algorithm, though one can use an arbitrarily complex function trading off interpretability.
>
> It is true that Figures 4-7 essentially contain all the information needed to judge about the optimizer’s tunability. However, a metric that is easy to compute, interpret and compare optimizers across tasks is crucial, for which we propose w-tunability. This is analogous to computing specific quantities like accuracy, FPR, TPR from the confusion matrix, even though a confusion matrix contains all the information (and is quite cumbersome to compare).
>
> The summary metric in Figure 2 provides a different interpretation: It reports a normalized performance i.e., the  normalized incumbent performance at iteration $k$. This doesn’t explicitly include information about the previous $k-1$ iterations (which is our central argument). Thus our proposed tunability metric provides more information than the summary statistics plot (figure 2).
>
> Due to this novelty, we argue that our setup does contribute to the qualitative understanding of the optimizers. In fact, it yields to a drastically different valuation of adaptive gradient methods than popular previous work (Wilson et al, Shah et al). You mention that our work is incremental to the work on benchmarking of optimizers. Can you please provide respective references?
>
> We have modified parts of our paper to reflect these arguments better.
>
> b. Using wall-clock time instead of HPO oracle calls:
> Our reason for using a number of configuration trials instead of a time budget is that measuring number of hyperparameter configuration searches required is more relevant to understand the optimizers’ dependence on the hyperparameters. However, we completely agree with you that computational budget is a relevant factor from the practitioner’s point of view, and added a discussion of this in Appendix E of the paper. As you rightfully point out, the adaptive optimizers tend to converge in fewer number of epochs, amplifying the results that favor Adam over the variants of SGD.
>
> References:
>
> Wilson, Ashia C., et al. "The marginal value of adaptive gradient methods in machine learning." Advances in Neural Information Processing Systems. 2017.
>
> Shah, Vatsal, Anastasios Kyrillidis, and Sujay Sanghavi. "Minimum norm solutions do not always generalize well for over-parameterized problems." arXiv preprint arXiv:1811.07055 (2018).
>
> Choi, Dami, et al. "On Empirical Comparisons of Optimizers for Deep Learning." arXiv preprint arXiv:1910.05446 (2019).

---

### Official Review · AnonReviewer2 · 2019-11-04
**Official Blind Review #2**

**Rating:** 3

**Review:**

This paper introduces a simple measure of tunability that allows to compare optimizers under varying resource constraints. The tunability of the optimizer is a weighted sum of best performance at a given budget. The authors found that in a setting with low budget for hyperparameter tuning, tuning only Adam optimizer’s learning rate is likely to be a very good choice; it doesn’t guarantee the best possible performance, but it is evidently the easiest to find well-performing hyperparameter configurations.

Comments:

The paper is easy to follow. The motivation of defining tunability of optimizer is a very interesting question, however, the study seems to preliminary and the conclusion is not quite convencing due to several reasons:

In section 3.2, to characterize its difficulties of finding best hyperparameters or tunability, the authors seem to try to connect the concept of “sharpness” of a minima in loss surface to the tunability of an optimizer, which is similar to comparing the loss landscape of minimums. However, while the authors made intuitive explanation about the tunability in section 2.2, I did not see the actual plot of the true hyperpaparameter loss surface of each optimizer to verify these intuitions. Can the author be more specific about the x-axis in the illustration 1.a and 1.b? If I understand correctly, they are not the number of trails.

In addition, the proposed stability metric seems not quite related with the above intuitions, as the illustrations (1.a and 1b) define the tunability to be the flatness of hyperparameter space around the best configurations, but the proposed definition is a weighted sum of the incumbents in terms of the HPO budgets.

The definition of the tuning budgets is not clear, is it the number of trials or the time/computation budgets? The authors seems interchangeably using “runs” and “iterations”, which makes the concept more confusable.

The authors further proposed three weighting schemes to emphasize the tunability of different stage of HPO. My concern is that is highly dependent  on the order of hyperparameter searched, which could impact the tunability significantly. For instance, in case of grid search HPO and 0.1 is the best learning rate, different search order such as [10, 1, 0.01, 0.1] and [0.1, 0.01, 1, 10] could results in dramatic different CPE and CPL.

My major concern is the hyperparameter distributions for each optimizer highly requires prior knowledge. A good prior of one optimizer could significantly affect the HPO cost or increase the tunability, i.e., the better understanding the optimizer, the less tuning cost. My major concern is that the authors assume the hyperparameters to be independent (section 3.2), which is not necessarily true. Actually hyperparameters are highly correlated, such as momentum, batch size and learning rate are correlated in terms of effective learning rate [1,2], so as weight decay and learning rate are [3], which means using non-zero momentum is equivalent to using large learning rate as long as the effective learning rate is the same. This could significantly increase the tunability of SGDM. Another concurrent submission [4] verified this equivalence and showed one can also just tune learning rate for SGDM. The assumption of independent hyperparameters might be fine for black box optimization or with the assumption that practitioners have no knowledge of the importance of each hyperparameter, then the tunability of the optimizer could be different based on the prior knowledge of hyperparameter and their correlations. But it is not rigorous enough to make the conclusion that Adam is easier to tune than SGD.


The author states their method to determine the priors by training each task specified in the DEEPOBS with a large number of hyperparameter samplings and retain the hyperparameters which resulted in performance within 20% of the best performance obtained. Could the authors be more specific on the hyperparameters searched? Is this process counted in the tunability measurement?

[1] Smith and Le, A Bayesian Perspective on Generalization and Stochastic Gradient Descent, https://arxiv.org/abs/1710.06451

[2] Smith et al, Don't Decay the Learning Rate, Increase the Batch Size, https://arxiv.org/abs/1711.00489

[3] van Laarhoven et al, L2 Regularization versus Batch and Weight Normalization, https://arxiv.org/abs/1706.05350

[4] Rethinking the Hyperparameters for Fine-tuning
 https://openreview.net/forum?id=B1g8VkHFPH


**Experience Assessment:**

I have published one or two papers in this area.

**Review Assessment: Checking Correctness Of Derivations And Theory:**

N/A

**Review Assessment: Checking Correctness Of Experiments:**

I assessed the sensibility of the experiments.

**Review Assessment: Thoroughness In Paper Reading:**

I read the paper at least twice and used my best judgement in assessing the paper.

---

> ### Author Response · Authors · 2019-11-06
> **Please add missing references**
>
> Dear reviewer,
>
> Thank you for your comment. Before we reply to the points you have made, we would kindly ask you to please add the references that are missing from your review.

---

> > ### Comment · AnonReviewer2 · 2019-11-06
> > **added the missing references**
> >
> > Sorry for missing the references.  Just updated my comments with them.

---

> ### Author Response · Authors · 2019-11-12
> **Reply to Reviewer #2 part 1**
>
> Thank you for your review of our work.
>
> The following are your concerns of our work:
> a. Prior distributions of hyperparameters
> b. Loss landscape plots and relation to tunability
> c. Importance of search order
> d. Details of the calibration procedure
>
> We address them as follows (in two parts):
>
> a. Prior distributions of hyperparameters:
> We envisage an optimizer not merely as update equations, but as the conjunction of the update equations, the hyperparameters, and distributions of those hyperparameters. Those distributions should be prescribed by the designers of the optimizer. This is crucial: For example, if we take Adam with LR between $10^1$ and $10^5$ and claim that Adam is less tunable than others, the evaluation is inherently faulty, as it doesn't capture where the mode of the distribution of LRs for which Adam is expected to work. These prescriptions are absent for the optimizers considered in the paper. Therefore, we define them from either mathematical reasoning (say learning rate is non-negative, $\beta_1, \beta_2$ in Adam are between (0, 1) and close to 1) or using the calibration step, where we determine those distributions by fitting on the configurations that yielded reasonably good results. We choose simple priors for their ease of estimation, though given enough computation, arbitrarily complex priors can be computed and used.
>
> We fail to see the explicit relationship between our work and the papers you have referenced. Specifically [1] only proposes that there is an optimal batch size that is dependent on the momentum parameter. We do not consider tuning the batch size, as we do not consider it a hyperparameter of the optimizer itself. [2] shows that instead of using LR decay schedule, increasing batch size has a similar effects on training, but results in faster training. [3] talks about the existence of an effective learning rate as a function of learning rate and the norm of the weights, and proposes that the optimal learning rate is inversely proportional to the weight decay parameter. This doesn’t, however, trivially lend itself to modeling priors. In summary, these papers show a complex interplay between the parameters giving rise to other notions, but not provide any methods to jointly model these hyperparameters. In the absence of such knowledge, we use our calibration procedure. The distributions we use are justified in section 3.2 in the paper. However, we accept the fact that a more complex distribution that might model the interaction between these hyperparameters might exist, and using that to sample for an HPO would be better.
>
> b. Loss landscape plots and relation to tunability:
> We show in figures 1.a, 1.b, as you rightly pointed out, the landscapes of loss function of the HPO objective as a function of the hypothetical hyperparameter $\theta$. There seems to be a misunderstanding of the purpose of figures 1.a, and 1.b.: These figures do not show what we try to measure, but they merely illustrate by example what properties we would like a tunability metric to have. We describe this in the beginning of Section 2. In Section 5, we explain why existing measures of tunability are unable to make the distinction between the cases in Figure 1a.
>
> We would like to emphasize that the point of Figures 1a, 1b is to illustrate the necessary properties that a proposed metric for tunability - it is not our intention to create such plots for our actual experiments. If you are interested in these nonetheless, a very recent publication by Asi & Duchi (2019) shows the plot of lr vs performance. In summary, they show that the sensitivity of SGD to stepsize choices, which converges only for a small range of stepsizes. AdamLR exhibits better robustness when tested on CIFAR10.

---

> > ### Author Response · Authors · 2019-11-12
> > **Reply to Reviewer #2 part 2**
> >
> >
> > c. Importance of search order:
> > You correctly observe that the tunability metric depends strongly on the order in which different hyperparameters are found, and can have a high variance because of that. In Section 4.1 we acknowledge that variance and explain how we quantify it: We simulate many reruns of the HPO with different random seeds and average the results across those different runs (i.e, for a budget of 4 across all these runs, what is the average CPE/U/L). As we use Random Search as HPO, the simulation of different seeds is achieved by shuffling the iterates. This shuffling and simulating multiple runs would have to replaced with running the code with different random seeds, if a bayesian HPO is used.
> >
> > d. Calibration details:
> > As Random Search algorithm requires priors to sample from, we calibrate the priors of the tunable parameters for each optimizer according to the calibration procedure described in Section 3.2. This process is only done to find areas of the hyperparameter that are plausible to have higher performance and does not directly contribute to the computed tunability metric.
> >
> > Additional details:
> > We use the term tuning budget to refer to the maximum number of times the HPO can call the oracle to get the function value. Each call to the oracle involves one retraining of the neural network itself and returning the appropriate validation metric; this oracle call is termed “iteration”. So we perform a total of “budget” number of “iterations”. We use the word “runs” to refer to one exhaustion of the above mentioned budget (to plot variance we simulate various “runs” with different random seeds). Each “run” can be understood as an independent execution of the HPO.
> >
> > We have included some of the points of this discussion in the paper.
> >
> > References:
> >
> > Asi, Hilal, and John C. Duchi. "The importance of better models in stochastic optimization." arXiv preprint arXiv:1903.08619 (2019).

---

### Public Comment · ~Supratik_Paul1 · 2019-10-28
**comments on the evaluation**

The choice of a log-normal distribution (table 2) over the learning rate seems a bit odd to me. Usually when performing grid search on the learning rate, points are exponentially spaced (even Schneider et al., 2019 do this). This suggests that for random search the learning rate should be sampled from an exponential distribution.

Also, the approach in Sec 3.2 of fitting to the top 20% injects bias in the evaluation in two ways. First, it gives undue preference to optimisers whose performance vs lr curve is closer to a log normal distribution, compared to optimisers which may exhibit more skewed curves, since the mode of the fitted distribution in the former will be closer to the max performance. Second, consider optimisers E and F from Fig 1b. The fitted log normal distribution for E will have lower variance than that of F. So by evaluating them based on samples from their respective fitted distributions, we lose information about the variability in performance and cannot make any inference of the sort mentioned in the figure. In an extreme scenario, E will always appear to be better than F.

I would be curious to know if the conclusions presented in the paper change if instead of sampling from the distribution fitted to the top 20%, you were to consider the full set of hyperparameters that were sampled initially.

---

> ### Author Response · Authors · 2019-11-02
> **Answers to questions on evaluation**
>
> Thank you for your interest in our work!
>
> We envisage an optimizer not merely as update equations, but as the conjunction of the update equations, the hyperparameters, and distributions of those hyperparameters. Those distributions should be prescribed by the designers of the optimizer. This is crucial: For example, if we take Adam with LR to be tuned from 10^1 to 10^5 and claim that Adam is less tunable than others, the evaluation is inherently faulty, as it doesn't capture where the mode of LRs for which Adam is expected to work. In the absence of those prescriptions, (like the optimizers we discuss in the paper), we define them from either mathematical reasoning (say learning rate is non-negative, \beta_1, \beta_2 in Adam are between (0, 1) and close to 1) or using the calibration step, where we determine those distributions by fitting on the configurations that yielded reasonably good results.
> We would first like to note that we do not take the top 20% best performing configurations for fitting the distributions in the calibration step, but consider only the configurations that are within 20% of the best performing configurations. This is to ensure that the fit is actually based on reasonably well performing configurations only. Crucially, we use the same fit for all datasets, which is in contrast to Choi et al (2019).
>
> a. Why we chose lognormal over exponential/other distribution families?
>
> Schneider et al (2019) use a logarithmic grid for learning-rate sampling, which requires an explicit choice of min and max of permissible range. Fixing those bounds can be tricky, as a hard maximum bound will preclude the possibility for any hyperparameter value to exceed that. Also, an exponential distribution has a mode of 0, for all values of \lambda, whereas a log-normal gives us the freedom to move the mode too. Inspecting the empirical distributions, we found that log-normal was a better fit than log-uniform for all optimizers. Since we do not discuss this in the current draft of the paper, we will add this information as soon as possible.
>
> b. Top 20% performers / optimizers E and F:
>
> As explained above, empirically, log-normal was the better fit for all the optimizers we considered, which is why we don’t think to have introduced an unfair bias.
> We emphasize that we all hyperparameter choices that result in a performance that are atleast 80% of the best attained performance from each task are taken, and MLE is computed on the combined top performers across all tasks; this means that there could be more than 0.2*#of tries that resulted in a "good enough" performance. To repeat, our priors are computed per optimizer and are made to be task independent (differing from Choi et al (2019)). Referring to optimizers E and F: that figure denotes the full \theta space for a single (hypothetical) problem. As you say, if the variance of log-normal for E is indeed low across all tasks, the problem of tuning is trivial; merely picking a value that previously worked is already close to optimum. Thus we argue that calibrating on all the tasks together, where the optimum of hyperparameters differs, is important and thus, in that case the variance of the lognormal fit to E will be higher.
>
> c. Do conclusions change with priors?:
>
> Our initial experiments (without calibration) showed similar trends and conclusions (as long as reasonable choices for the priors of the hyperparameters are made), meaning that the effect of the calibration step is likely small for the optimizers we considered. However, we argue that the step is necessary to comply with our described scenario of having a pre-defined distribution to sample from.
>
> Refs:
> Dami Choi, Christopher J Shallue, Zachary Nado, Jaehoon Lee, Chris J Maddison, and George EDahl. On empirical comparisons of optimizers for deep learning.arXiv preprint arXiv:1910.05446,2019

---

### Public Comment · ~Carmen_Amo_Alonso1 · 2019-11-02
**comments on evaluations**

The authors did not provide a fair comparison between adaptive and non-adaptive methods, they did not consider step size scheduling for SGD methods, according to the prior work (see [1] section 4.1, and section 5.1 of [2]) step size scheduling is the parameter impacting the performance the most, therefore the results of this paper are wrong and flawed. This perhaps also explains why the results of this paper do not match the prior work [1, 2].


[1] Wilson, Ashia C., et al. "The marginal value of adaptive gradient methods in machine learning." Advances in Neural Information Processing Systems. 2017.
[2] Shah, V., Kyrillidis, A., & Sanghavi, S. (2018). Minimum norm solutions do not always generalize well for over-parameterized problems. arXiv preprint arXiv:1811.07055.

---

> ### Author Response · Authors · 2019-11-04
> **Reply to comments on evaluation**
>
> It is true that a learning rate schedule can have a profound impact on the performance of an optimizer. The same is true for momentum and weight decay, which results in our study considering 5 different SGD variants in total, of which the most flexible one has 3 optimizable hyperparameters. Integrating a learning rate schedule would introduce at least one more and further increase the combinatorial complexity of the search space. For this reason, it is possible, but certainly not guaranteed, that a learning rate schedule would improve the tunability of SGD (although it almost certainly improves the best achievable performance of SGD), especially since it can be applied to Adam as well.
>
> Initially, we did not consider learning rate schedules since it was missing from the DeepOBS benchmark as well (Schneider et al.). Due to its importance, we are planning to include results of optimizers using a learning rate schedule in the final version of the paper (in main sections or appendix). However, it is important to understand that the value of our paper is not dependent on the results of the optimizers we considered. Our main goal is to improve the evaluation protocols for optimizers by introducing a fair hyperparameter search protocol largely independent of human input. On that note, our paper is not the only one that observes results different from what Wilson et al. reported: Choi et al. demonstrate that Wilson et al.’s result strongly depend on their hyperparameter search method, a similar argument as in our paper.
>
> References:
>
> Dami Choi, Christopher J Shallue, Zachary Nado, Jaehoon Lee, Chris J Maddison, and George EDahl. On empirical comparisons of optimizers for deep learning.arXiv preprint arXiv:1910.05446,2019
>
> Frank Schneider, Lukas Balles, and Philipp Hennig. DeepOBS: A deep learning optimizer benchmark suite. In International Conference on Learning Representations, 2019. URL https://openreview.net/forum?id=rJg6ssC5Y7.

---

### Public Comment · ~Carmen_Amo_Alonso1 · 2019-11-02
**comments on evaluation**

At the end of section 3.1, authors choose to stop training when the validation loss plateaus for more than 2 epochs or if the number of epochs exceeds the predetermined maximum number. This however provides unfair evaluation since the performance of an optimizer can get possibly better after 2 epochs, as suggested in [1] section 4, the authors could allocate a pre-specified budget on the number of epochs used for training each model and report the best peak performance on the development set by the end of the fixed epoch budget.

[1] Wilson, Ashia C., et al. "The marginal value of adaptive gradient methods in machine learning." Advances in Neural Information Processing Systems. 2017.

---

> ### Author Response · Authors · 2019-11-04
> **We argue that validation loss-based stopping is the best option**
>
> Your considerations are reasonable and have been made by us as well. However, we ultimately decided that early stopping is the fairest option. It is true that the performance of an optimizer may become better after 2 or more epochs of no improvement. We did consider the alternative approach that you suggest, but we decided against it for the following reasons: Setting a fixed model and dataset specific number of epochs is against the spirit of our paper, because convergence speed is an important characteristic of an optimizer. In fact, we report a substantially faster convergence of AdamLR compared to SGDMC.

---

### Public Comment · ~Jongho_Kim1 · 2019-11-02
**comments on the model and evaluations**

I am concerned about the proposed definition of tunability. The authors provide a definition of tunability where one can give more weight to “early stage” or “late-stage” of hyperparameter tuning, however, during hyperparameter tuning, selection of different hyperparameters are independent of each other and is done with random search, considering “early stage” or “late-stage” does not make sense, since these variables are independently chosen of each other. The authors, therefore, do not provide a definition useful for the evaluation of neural net optimizers. In practice, one still will report the best hyperparameters over the given budget.


Paper [1] provides standard benchmarks with a different set of models and architectures, they discuss the hyperparameters tuning in section 2.4 thoroughly. Given prior work in this domain [1][2][3], the authors do not provide any novel method. The definition is not well designed, and the work does not provide any useful contribution that will ease the rigor of neural net optimizer evaluation. The authors additionally do not provide any results on the generalization capability of the optimizers which is an important aspect when it comes to optimizer benchmarking as done in [2].


As other reviewers point out, the learning rate schedule is critical to get reasonable performance in the considered optimizers and the provided comparisons are not fair.


Figure 1 and given arguments in 2.2,  are not valid for the loss surface of neural networks, which have a nonconvex loss. Therefore in practice, one would report the best hyper-parameter over the given budget.





[1] Frank Schneider, Lukas Balles, and Philipp Hennig. DeepOBS: A deep learning optimizer benchmark suite. In International Conference on Learning Representations, 2019. URL https://openreview.net/forum?id=rJg6ssC5Y7.
[2] Wilson, Ashia C., et al. "The marginal value of adaptive gradient methods in machine learning." Advances in Neural Information Processing Systems. 2017.
[3] Shah, V., Kyrillidis, A., & Sanghavi, S. (2018). Minimum norm solutions do not always generalize well for over-parameterized problems. arXiv preprint arXiv:1811.07055.

---

> ### Author Response · Authors · 2019-11-04
> **Reply to comments on our evaluation and considered models**
>
> Thank you for your comment!
>
> Our understanding is that you have the following concerns:
>
> 1. The definition of tunability boils down to reporting the best hyperparameters for a given budget.
> 2. Our analysis provides no novelty considering the previous work in this research area.
> 3. Our paper does not provide theoretical results on the generalization abilities of optimizers.
> 4. We do not consider learning rate schedules.
> 5. Figure 1 and arguments are not valid due to non-convexity of neural networks.
>
> We would like to address them one by one.
>
> 1.
> Your understanding is almost correct. At each step t, we measure the performance of the best hyperparameter configuration up to that point, i.e., for a budget t. We could have chosen particular “t”s to represent small, medium, and large budgets, but this choice would always be arbitrary and can yield different results depending on the choice. To mitigate this issue, we take a weighted average over all steps, where we put more emphasis on the early steps in one case, and more emphasis on the later steps in the other case. This is what “early stage” and “late stage” refers to. These phrases do NOT refer to the specific configurations found in different stages of the hyperparameter tuning method, which would indeed not make sense, as you point out correctly. Instead, the different weighting schemes account for different opinions of what one might value in an optimizer: Getting to good performances with little tuning effort (Cumulative Performance Early) or achieving best performances with a lot of tuning effort (Cumulutative Performance Late). It's to be noted that merely reporting the best hyperparameters ignores the hyperparameter tuning budget itself, and our metrics of CPU/E/L encode this information.
>
> 2.
> We extensively discuss all three works you mention in the related work section 5. We argue carefully how our paper differs from each of these in important ways.
>
> 3.
> We indeed do not conduct any theoretical analysis of existing optimizers, but we never claim nor intend to do so. While theoretical analyses as conducted by Wilson et al. are interesting and valuable in itself, our paper has an empirical focus and addresses important shortcomings of the experimental setup from previous studies (discussed in Section 5). We are not the only work addressing this important issue: For example, Choi et al. provide evidence that the results obtained by Wilson et al. are only due to their specific hyperparameter search protocol.
>
> 4.
> Regarding learning rate schedules, please refer to the reply to the comment of Carmen Amo Alonso addressing the same issue.
>
> 5.
> Figure 1 does not show the loss as a function of the model parameters (which is indeed non-convex in most deep neural networks), but as a function of some hyperparameter, e.g., the learning rate. While this function is not necessarily convex either, empirically it often behaves approximately convex (see e.g. Goodfellow et al. Figure 11.1, Schneider et al. Figure 3), so that our figures are not entirely unrealistic. Regardless of this point, our argumentation makes no convexity assumption at any point: The figures show convex functions for the sake of simpler presentation, but our arguments could be visualized with non-convex functions as well.
>
> Dami Choi, Christopher J Shallue, Zachary Nado, Jaehoon Lee, Chris J Maddison, and George EDahl. On empirical comparisons of optimizers for deep learning.arXiv preprint arXiv:1910.05446,2019
>
> Frank Schneider, Lukas Balles, and Philipp Hennig. DeepOBS: A deep learning optimizer benchmark suite. In International Conference on Learning Representations, 2019. URL https://openreview.net/forum?id=rJg6ssC5Y7.
>
> Ian Goodfellow, Yoshua Bengio, Aaron Courville. Deep Learning. In MIT press, 2019. URL https://www.deeplearningbook.org/contents/guidelines.html

---

### Public Comment · ~Abhyuday_Jagannatha4 · 2019-11-04
**experimental evaluation**

I'm not an expert on this literature but I think it's important to have practical evaluations like this, but I also think that many of their conclusions are just common sense, e.g., page 2, "[a]lthough a well-tuned SGD variant is able to reach the top performance in some cases, [overall results] favor adaptive gradient methods", page 3 "[t]hus, the difference of the two interpretations of tunability lies whether one values results from late stages [...] more than results from early stages", the discussion in that page, etc.

Now, for the definition of w-tunability - I'm sorry, but this doesn't make too much sense to me. What is a "budget of K runs"? Different algorithms could have wildly different cost per "run" (think one update of SGD vs. one update of full-matrix AdaGrad), so what gives then? Why would this be relevant from the point of view of practical optimization?

I don't think there's any mention of FLOPS/iteration in the paper. Are all algorithms they test really equivalent in terms of FLOPS/run? If not, is the notion of a "K-budget" really relevant? Wouldn't it make more sense to define "tunability" in terms of a budget of FLOPS (or processor time elapsed) instead?

---

### Public Comment · ~John_Duchi1 · 2019-11-05
**Some related work**

Hi authors-

Broadly, the goal of understanding the robustness of an optimization method is important, and your discussion that it is not an optimizer with the best performance for precisely the right hyperparameter settings that is the best, but instead one that performs broadly well, is valuable.

With my student Hilal, we have a series of works (https://www.pnas.org/content/early/2019/10/29/1908018116.abstract, https://epubs.siam.org/doi/abs/10.1137/18M1230323, http://web.stanford.edu/~jduchi/projects/AsiDu19-bregman.pdf, with arXiv versions https://arxiv.org/abs/1903.08619, https://arxiv.org/abs/1810.05633, and talks, e.g. https://simons.berkeley.edu/talks/tbd-28) that both aim to build robust methods and "we argue for a different type of experimental evidence in evaluating stochastic optimization methods, where one jointly evaluates convergence speed and sensitivity of the methods." While these papers do not explicitly define a single metric for optimizer performance, one of their points is to show both that it is possible to build robust methods and to demonstrate visualization strategies for comparing sensitivity/robustness of the algorithms.

It seems that acknowledging this prior line of work would be appropriate.

---

> ### Author Response · Authors · 2019-11-08
> **Thank you pointing out relevant literature**
>
> Hi,
>
> Thank you for pointing out relevant literature that we missed out. We will investigate and discuss them in the future versions of our work.

---

### Decision · Program_Chairs · 2019-12-19

**Decision:**

Reject

**Comment:**

The paper proposed a new metric to define the quality of optimizers as a weighted average of the scores reached after a certain number of hyperparameters have been tested.

While reviewers (and myself) understood the need to better be able to compare optimizers, they failed to be convinced by the proposed solutions. In particular (setting aside several complaints of the reviewers with which I disagree), by defining a very versatile metric, this paper lacks a strong conclusion as the ranking of optimizers would clearly depend on the instantiation of that metric.

Although that is to be expected, by the very behaviour of these optimizers, it makes it unclear what the added value of the metric is. As one reviewer pointed out,  all the points made could have been similarly made with other, more common plots.

Ultimately, it wasn't clear to me what the paper was trying to achieve beyond defining a mathematical formula encompassing all "standard" evaluation metric, which I unfortunately see of limited value.